# Scutellarein Suppresses the Production of ROS and Inflammatory Mediators of LPS-Activated Bronchial Epithelial Cells and Attenuates Acute Lung Injury in Mice

**DOI:** 10.3390/antiox13060710

**Published:** 2024-06-12

**Authors:** Ximeng Li, Xiaoyu Zhang, Yuan Kang, Min Cai, Jingjing Yan, Chenchen Zang, Yuan Gao, Yun Qi

**Affiliations:** State Key Laboratory of Bioactive Substance and Function of Natural Medicines, Institute of Medicinal Plant Development, Chinese Academy of Medical Sciences & Peking Union Medical College, Beijing 100193, China; liximeng@implad.ac.cn (X.L.); zhangxiaoyu@etinpro.com (X.Z.); kouenn1999@gmail.com (Y.K.); caimin@implad.ac.cn (M.C.); yanjingjing@implad.ac.cn (J.Y.); arclyn74@gmail.com (C.Z.)

**Keywords:** scutellarein, reactive oxygen species (ROS), inflammation, bronchial epithelial cells, lipopolysaccharide (LPS), acute lung injury (ALI)

## Abstract

Scutellarein is a key active constituent present in many plants, especially in *Scutellaria baicalensis* Georgi and *Erigeron breviscapus* (vant.) Hand-Mazz which possesses both anti-inflammatory and anti-oxidative activities. It also is the metabolite of scutellarin, with the ability to relieve LPS-induced acute lung injury (ALI), strongly suggesting that scutellarein could suppress respiratory inflammation. The present study aimed to investigate the effects of scutellarein on lung inflammation by using LPS-activated BEAS-2B cells (a human bronchial epithelial cell line) and LPS-induced ALI mice. The results showed that scutellarein could reduce intracellular reactive oxygen species (ROS) accumulation through inhibiting the activation of NADPH oxidases, markedly downregulating the transcription and translation of pro-inflammatory cytokines, including interleukin-6 (IL-6), C-C motif chemokine ligand 2 (CCL2), and C-X-C motif chemokine ligand (CXCL) 8 in LPS-activated BEAS-2B cells. The mechanism study revealed that it suppressed the phosphorylation and degradation of IκBα, consequently hindering the translocation of p65 from the cytoplasm to the nucleus and its subsequent binding to DNA, thereby decreasing NF-κB-regulated gene transcription. Notably, scutellarein had no impact on the activation of AP-1 signaling. In LPS-induced ALI mice, scutellarein significantly decreased IL-6, CCL2, and tumor necrosis factor-α (TNF-α) levels in the bronchoalveolar lavage fluid, attenuated lung injury, and inhibited neutrophil infiltration. Our findings suggest that scutellarein may be a beneficial agent for the treatment of infectious pneumonia by virtue of its anti-oxidative and anti-inflammatory activities.

## 1. Introduction

Acute lung injury (ALI) is a severe respiratory system disease triggered by endogenous or exogenous pathogenic factors. It is clinically shown as overwhelming lung inflammation, diffuse alveolar damage, pulmonary edema, and progressive hypoxemia [1]. Currently, no effective therapeutic modalities are available for the treatment of ALI. Recent evidence has demonstrated that suppressing both the release of pro-inflammatory mediators and the formation of reactive oxygen species (ROS) as early as possible may be a promising therapy [2]. As we know, gram-negative bacterial infection stands as a key causative factor of ALI, and lipopolysaccharide (LPS), the major component of the outer membranes of gram-negative bacteria, can provoke the excess production of pro-inflammatory mediators and ROS of immune cells and non-immune cells, thus instigating lung injury [3,4]. Therefore, it becomes a practical inducer for ALI-related models in vitro and in vivo [5], such as LPS-activated macrophages or epithelial cells in the lung [6,7] and LPS-induced ALI mice [8].

Upon interacting with LPS, toll-like receptor 4 on airway epithelial cells initiates a sequence of inflammatory reactions via the myeloid differentiation factor 88 (MyD88)-dependent pathway. This process further triggers two well-established inflammatory pathways: NF-κB and AP-1 signaling [9,10]. The activation of either of them leads to the transcription of pro-inflammatory genes, such as interleukin-6 (IL-6), C-C motif chemokine ligand 2 (CCL2), C-X-C motif chemokine ligand (CXCL) 8, and tumor necrosis factor-α (TNF-α) [11,12]. Although inflammation assists in pathogen elimination, excessive pulmonary inflammatory responses can disrupt epithelial integrity and elevate alveolar-capillary barrier permeability, thus resulting in lung parenchymal injury and ALI [13]. Hence, suppressing lung inflammation holds promise as a viable therapeutic approach for ALI treatment.

Scutellarein (synonyms: 6-hydroxyapigenin; CAS^#^529-53-3; Figure 1) is one of the active constituents present in both the aerial parts and roots of *Scutellaria baicalensis* Georgi [14,15]. The aerial parts, including the stems and leaves, are consumed as a folk “Huangqin tea” in northern and southwest China [16]. The roots are widely used as a traditional Chinese medicine for the treatment of respiratory infections [17,18,19]. Scutellarein is also the metabolite of scutellarin (CAS^#^27740-01-8), which is capable of relieving LPS-induced ALI [20,21], and scutellarein can suppress LPS-induced inflammatory responses in mouse peritoneal macrophages (RAW264.7) [22]. Therefore, we surmised that scutellarein should be an active constituent for the treatment of infectious respiratory inflammation. As we know, the lungs present the largest epithelial surface for interactions with infectious pathogens and toxins that cause acute lung injury. Epithelial cells can produce a spectrum of pro-inflammatory cytokines, chemokines, as well as ROS [23,24]. In the present study, we investigated the effects of scutellarein on ROS and the inflammatory mediators of LPS-activated BEAS-2B cells (a human bronchial epithelial cell line) and LPS-induced ALI mice.

## 2. Materials and Methods

### 2.1. Reagents

Scutellarein (Batch^#^MUST-22061007, Purity: 99.79%) was purchased from Chengdu Must Bio-technology Co. (Chengdu, China). The bronchial epithelial cell growth medium (BEGM) kit (Cat^#^CC-3170) was from Lonza (Visp, Switzerland). LPS (Cat^#^L4130) and lucigenin (Cat^#^M8010) were from Sigma-Aldrich (St. Louis, MO, USA). L-012 (Cat^#^120-04891) was from Wako Chemicals (Tokyo, Japan). MitoSOX™ mitochondrial superoxide indicator (Cat#M36008) was from Thermo Fisher Scientific Inc. (Waltham, MA, USA). The ELISA kits for human IL-6 (Cat^#^430504), CCL2 (Cat^#^438804), CXCL8 (Cat^#^431504), mouse IL-6 (Cat^#^431301), TNF-α (Cat^#^430901), and CCL2 (Cat^#^432701) were provided by BioLegend Co. (San Diego, CA, USA). The TRIzol reagent (Cat^#^15596026) was from Invitrogen Co. (Carlsbad, CA, USA), and the M-MuLV first strand cDNA synthesis kit (Cat^#^B532435-0100) was obtained from Sangon Biotech Co., Ltd. (Shanghai, China). The antibodies against GAPDH (Cat^#^AC033), histone H3 (Cat^#^A2348), ERK (Cat^#^A10613), and HRP goat anti-mouse or rabbit IgG (H+L) (Cat^#^AS003 and AS014) were from ABclonal Biotech Co. (Wuhan, China). The antibodies against NF-κB p65 (Cat^#^4764), JNK (Cat^#^9258), p-JNK (Cat^#^9251), p-ERK (Cat^#^4370), p38 (Cat^#^9212), and p-p38 (Cat^#^9215) were from Cell Signaling Technology (Danvers, CO, USA). The nuclear and cytoplasmic protein extraction kit (Cat^#^P0028), BCA protein assay kit (Cat^#^P0012), and antibodies against IκBα (Cat^#^AF1282) and p-IκBα (Cat^#^AF1870) were obtained from the Beyotime Institute of Biotechnology (Haimen, China). The TransAM™ NF-κB p65 transcription factor assay kit (Cat^#^40096) was from Active Motif, Inc. (Carlsbad, CA, USA).

### 2.2. Cells

The BEAS-2B cell line was obtained from the China Center for Type Culture Collection and cultured in BEGM culture medium in a humidified incubator set at 37 °C and 5.0% CO_2_. Based on the growth curve analysis, the cells were seeded at a density of 1.6 × 10^5^ cells/mL.

### 2.3. Animals and Ethical Statement

Male ICR mice (SPF grade, 6–8 weeks, 20 ± 2 g) were obtained from Vital River Experimental Animal Services [Beijing, China; license No. SCXK (Beijing) 2021-0006, approval date: 26 July 2021]. Prior to the experiments, the mice were acclimated to the environment for 3 days, with free access to food and water, and maintained on a 12 h light/dark schedule. All experiments were approved by the Care and Use Committee of the Institute of Medicinal Plant Development (IMPLAD) of the Chinese Academy of Medical Sciences (approval No. IMPLAD-2022-1109, approval date: 15 November 2022) and carried out in accordance with the Guide for the Care and Use of Laboratory Animals and the recommendations in the ARRIVE guidelines. The anesthetic drug (isoflurane) and all other necessary measures were used to ameliorate any suffering for the animals.

### 2.4. Cytotoxicity Assay

The MTT assay was employed to assess the cell viability. BEAS-2B cells (5 × 10^5^ cells per well) were treated with scutellarein at final concentrations of 12.5–200 μM for 20 h and then exposed to a 0.5% MTT solution for an additional 4 h. Subsequently, the supernatant was aspirated, and 100 μL of DMSO was introduced to each well to dissolve the formed formazan crystals. The optical density at 540_nm_ was then recorded.

### 2.5. Measurement of the Intracellular ROS

The intracellular ROS was measured using the chemiluminescence probe L-012 as previously described [25]. In a white 96-well cell culture plate, BEAS-2B cells (5 × 10^5^ cells per well) were pretreated with scutellarein for 1 h and then stimulated with LPS (250 ng/mL) for 24 h. After washing the cells twice with D-Hank’s, an L-012 probe (100 μM) was added, followed by incubation at 37 °C for 30 min. The chemiluminescence was detected using a microplate counter (MicroBeta2, Perkin Elmer, Waltham, MA, USA).

### 2.6. Measurement of the Intracellular NADPH Oxidase Activity

In a 6-well cell culture plate, BEAS-2B cells (8 × 10^6^ cells per well) were pretreated with scutellarein for 1 h and then stimulated with LPS (250 ng/mL) for 6 h. The cells were washed twice with PBS and then sonicated on ice in a 50 mM KH_2_PO_4_ buffer (pH 7.0) containing 1 mM EDTA and a protease inhibitor cocktail. The supernatant of the cell lysate was collected after centrifugation at 13,000× *g* at 4 °C for 5 min. Then, 20 μL of the supernatant was added into a white 96-well plate and incubated with 80 μL of lucigenin at 25 °C in the dark for 5 min. After the addition of 1 mM NADPH, the chemiluminescence was detected immediately using a microplate counter (MicroBeta2, Perkin Elmer, USA) [25]. The protein concentration was standardized using the BCA protein assay kit.

### 2.7. Measurement of the Mitochondrial ROS

In a black 96-well cell culture plate, BEAS-2B cells (5 × 10^5^ cells per well) were pretreated with scutellarein for 1 h and then stimulated with LPS (250 ng/mL) for 24 h. After washing the cells twice with D-Hank’s, MitoSOX red mitochondrial superoxide indicator (5 μM) was added, followed by incubation at 37 °C for 30 min. After washing the cells twice with D-Hank’s, the fluorescence was read at λex 510_nm/_λem 590_nm_ using a fluorescence microplate analyzer (Fluoroskan Ascent FL, Thermo Fisher Scientific, Waltham, MA, USA) [25].

### 2.8. Measurement of the Pro-Inflammatory Cytokines

The BEAS-2B cells (5 × 10^5^ cells per well) were pretreated with scutellarein (12.5–50 μM) for 1 h and then stimulated with LPS (250 ng/mL) for 24 h. The supernatants were collected and analyzed for IL-6, CCL2, and CXCL8 using human ELISA kits. The concentrations of these cytokines were calculated by comparing their respective standard curves.

### 2.9. Extraction of the RNA and Performing Real-Time Quantitative PCR (RT-qPCR)

RT-qPCR was utilized to assess the mRNA levels of the pro-inflammatory cytokines. The BEAS-2B cells were pretreated with scutellarein (12.5–50 μM) for 1 h and then stimulated with LPS (250 ng/mL) for 4 h. The Trizol reagent was used to extract the total mRNA from the cells, followed by performing reverse transcription reactions with the M-MuLV first-strand cDNA synthesis kit. A BIOER fluorescent quantitative detection system was used for the RT-qPCR analyses with the following protocol: 30 s at 95 °C, 40 cycles of 5 s at 95 °C, and 30 s at 62 °C. The relative expression of the pro-inflammatory genes was analyzed using the comparative Ct method (2^−ΔΔCt^). GAPDH served as the housekeeping gene. All primer sequences are listed in Table 1.

### 2.10. Western Blot Assay

In 6-well culture plates, BEAS-2B cells (8 × 10^6^ cells per well) were pretreated with scutellarein at concentrations of 12.5, 25, and 50 μM for 1 h and then stimulated with LPS (250 ng/mL) for varying time points (IκBα, p-IκBα, and p65: 30 min; p-ERK, p-JNK, and p-p38: 15 min). Mammalian protein extraction kits were used for extracting the total proteins. For the isolation of the nuclear and cytoplasmic proteins, nuclear and cytoplasmic protein extraction kits were utilized. The protein concentrations were quantified utilizing the BCA protein assay kit.

For the Western blot analysis, protein samples (30 µg/lane) were loaded, separated using 10–12% sodium dodecyl sulfate-polyacrylamide gel electrophoresis (SDS-PAGE), and subsequently transferred onto PVDF membranes. Then, the membranes were blocked with 5% non-fat milk dissolved in TBST and incubated at 25 °C for 2 h. Subsequently, the membranes were incubated with various primary antibodies at 4 °C overnight. Following three washes using TBST, the membranes were then incubated with the corresponding secondary antibodies for 1 h at 25 °C. After performing additional washes, the protein bands were visualized using an ECL Western blot kit and captured using a ChemiScope capture system.

### 2.11. LPS-Induced ALI Mice

The LPS-induced ALI mouse model was conducted in accordance with a previously established protocol [26], with slight modifications. In brief, ICR mice were randomly divided into 5 groups (11 mice per group): a negative control group, an LPS model group (2 μg/mouse of LPS), and 3 scutellarein treatment groups (25 mg/kg, 50 mg/kg, or 100 mg/kg of scutellarein + 2 μg/mouse of LPS). The dosages of scutellarein were determined using pilot studies. The mice were intraperitoneally injected with scutellarein dissolved in normal saline containing 5% DMSO and 2% Tween-80 [27]. The mice in the negative control and LPS model groups received an equal volume of vehicles. Thirty minutes later, the mice were anesthetized using 2% isoflurane, and ALI was induced via the intratracheal delivery of LPS (2 μg/20 μL/mouse) using a mouse microsprayer aerosolizer (BioJane, Shanghai, China). The mice in the negative control group were administered an equivalent volume of normal saline. Six hours after LPS administration, the mice were anesthetized and humanely euthanized for sample collection.

Three mice from each group were randomly chosen for pathological examination. The whole lungs were fixed in 4% paraformaldehyde, embedded in paraffin, sectioned into 3 μm slices, and stained with hematoxylin and eosin (H and E) for the analysis using a digital slide scanner (3Dhistech, Pannoramic MIDI, Budapest, Hungary). To quantify the severity of the lung injury via histology, six random high-power fields (200×) per mouse lung were analyzed blindly and scored based on criteria by Smith Kendra M. MD [28]. Edema, alveolar and interstitial inflammation, hemorrhage, atelectasis, and hyaline membrane formation were each scored on a 0–4 scale: no injury = 0; injury in 25% of the field = 1; injury in 50% of the field = 2; injury in 75% of the field = 3; and injury throughout = 4. The total lung injury score is the sum of these individual scores.

The remaining mice were used for bronchoalveolar lavage fluid (BALF) collection, which was obtained via intratracheal intubation, followed by lavaging the lungs with 1 mL of ice-cold PBS five times. Cytokine concentrations in the BALF were determined using commercial ELISA kits.

### 2.12. Statistical Analysis

GraphPad Prism (version 7.0) was used for all statistical analyses. The Shapiro–Wilk test was used for assessing the normality. For the normally distributed data, unpaired Student’s *t*-tests were employed to compare two groups, while a one-way ANOVA with Tukey’s post hoc analysis was utilized for comparing multiple groups. For the data not normally distributed, the Mann–Whitney test was used for comparing two groups, and the Kruskal–Wallis test with Dunn’s post hoc analysis was used for comparing multiple groups. The data were reported as the mean ± SD from a representative experiment. All the experiments were repeated at least three times with the same pattern of results. Lastly, *p* < 0.05 was considered significant.

## 3. Results

### 3.1. Scutellarein Exerts Anti-Oxidative Activity in LPS-Activated BEAS-2B Cells

To rule out the non-selective inhibition of scutellarein caused by cytotoxicity, we first investigated its effect on the viability of BEAS-2B cells. The results showed that scutellarein did not affect the cell viability when the concentrations were no more than 50 μM (Figure 2A). As shown in Figure 2B, scutellarein (12.5–50 μM) could significantly reduce LPS-induced intracellular ROS accumulation. Given the critical role of NADPH oxidase in generating intracellular ROS, we next determined the effect of scutellarein on the activity of NADPH oxidase. As shown in Figure 2C, scutellarein markedly inhibited the activation of NADPH oxidase in LPS-activated BEAS-2B cells. However, scutellarein did not decrease mitochondrial ROS production at the non-toxic concentrations (Figure 2D).

### 3.2. Scutellarein Decreases the Supernatant Pro-Inflammatory Mediator Production in LPS-Activated BEAS-2B Cells

To evaluate the anti-inflammatory activity of scutellarein, we first determined its actions on the supernatant pro-inflammatory mediators in LPS-stimulated BEAS-2B cells at the non-toxic concentrations (12.5–50 μM). As shown in Figure 3, scutellarein significantly decreased supernatant IL-6, CCL2, and CXCL8, with the IC_50_ values of 36.7 ± 11.0 μM, 20.2 ± 5.6 μM, and 13.2 ± 2.3 μM, respectively, showing potent anti-inflammatory activity.

### 3.3. Scutellarein Decreases the mRNA Levels of Pro-Inflammatory Mediators in LPS-Stimulated BEAS-2B Cells

Given the suppressive effect of scutellarein on IL-6, CCL2, and CXCL8 released by LPS-activated BEAS-2B cells, we proceeded to investigate its effects on the mRNA expression of these pro-inflammatory mediators. The RT-qPCR analyses revealed that scutellarein could potently lower the mRNA levels of IL-6, CCL2, and CXCL8 in LPS-activated BEAS-2B cells (Figure 4), indicating that its inhibitory effect on supernatant pro-inflammatory mediators resulted from its capacity to suppress their transcription.

### 3.4. Scutellarein Suppresses NF-κB Activation via Inhibiting IκBα Degradation in LPS-Stimulated BEAS-2B Cells

NF-κB (p65/p50 heterodimers) holds a crucial role in orchestrating inflammatory responses by regulating the transcription of pro-inflammatory genes. We next explored the influence of scutellarein on the key events within the NF-κB pathway. As shown in Figure 5A–C, scutellarein hindered the LPS-caused translocation of p65 from the cytoplasm to the nucleus and inhibited p65–DNA binding. Since phosphorylated and degraded IκBα leads to the liberation of p65/p50 heterodimers, we proceeded to assess scutellarein’s effects on IκBα and p-IκBα expression levels. Our findings demonstrated that following LPS stimulation for 30 min, IκBα underwent phosphorylation and degradation in BEAS-2B cells, while scutellarein markedly inhibited these changes in a concentration-dependent manner (Figure 5D,E).

### 3.5. Scutellarein Does Not Affect the Phosphorylation of Mitogen-Activated Protein Kinases (MAPKs) in LPS-Stimulated BEAS-2B Cells

In addition to NF-κB, AP-1 also governs inflammatory genes during LPS-triggered inflammatory responses. The activation of AP-1 is mediated by the MAPK family, including ERK, JNK, and p38. As demonstrated in Figure 6, the levels of p-ERK, p-JNK, and p-p38 notably increased when the cells were exposed to LPS for 15 min, while scutellarein did not influence their phosphorylation levels, indicating that scutellarein curbed LPS-induced inflammation in BEAS-2B cells independent of the MAPKs/AP-1 pathway.

### 3.6. Scutellarein Ameliorates LPS-Induced ALI In Vivo

Given the fact that ALI is characterized by the release of inflammatory cytokines in the lung, we also determined the in vivo anti-inflammatory impact of scutellarein on LPS-induced ALI mice. As illustrated in Figure 7A–C, the levels of IL-6, CCL2, and TNF-α in BALF were robustly increased 6 h after LPS administration via intratracheal delivery, while a single administration of scutellarein (25–100 mg/kg, *i.p.*) resulted in a notable decrease in their levels in the BALF. Moreover, LPS also caused significant pathological changes in the pulmonary tissues, including septal thickening, interstitial edema, alveolar hemorrhage, and pronounced neutrophilic infiltration, while scutellarein was able to relieve these pathological changes (Figure 7D,E and Figure 8). These findings demonstrate the efficacy of scutellarein in attenuating lung inflammatory responses.

## 4. Discussion

Flavonoids, a class of phenolic phytochemicals widely distributed in edible plants (e.g., fruits and vegetables), have garnered considerable attention due to their advantageous physiological effects and notable hypotoxicity [29]. Among them, scutellarein stands out as an effective component derived from many plants, especially in *Scutellaria baicalensis* Georgi [30] and *Erigeron breviscapus* (vant.) Hand-Mazz [31], which possesses both anti-inflammatory and anti-oxidative activities [32,33,34]. The chemical structure of scutellarein evolved from the skeleton of 2-phenyl chromogenic ketone (C6-C3-C6) and contains multiple hydroxyl groups, which are crucial for its activity [35]. Notably, as an active metabolite of scutellarin in vivo, scutellarein possesses enhanced solubility, bioactivity, and bioavailability compared to scutellarin [36]. The present study provides evidence supporting scutellarein as a promising anti-oxidative and anti-inflammatory agent for infectious pneumonia.

Oxidative stress and the resulting inflammation are essential pathological processes in ALI. LPS is widely used to increase the production of ROS and has become one of the most important causes of ALI. Our results demonstrated that scutellarein markedly reduced the LPS-induced intracellular ROS accumulation of human bronchial epithelial cells (BEAS-2B) (Figure 2B). ROS are mainly generated from two main sources: the mitochondrial electron transport chain and membrane enzymatic complexes, known as NADPH oxidases. Lung epithelial cells produce ROS mainly from NADPH oxidase 1 (NOX1) [24]. Scutellarein could significantly inhibit the activity of NADPH oxidases but did not decrease mitochondrial ROS (Figure 2B–D). Given that ROS can also be directly scavenged, we further assayed the ability of scutellarein to remove superoxide anion, the key member of ROS and the typical primary product of the electron transfer reaction catalyzed by NADPH oxidases [37,38]. As a result, scutellarein barely scavenged superoxide anions at 50 μM (IC_50_ > 400 μM; Appendix A). These findings indicate that scutellarein is an inhibitor of NADPH oxidases but is not a superoxide anion scavenger.

In fact, although the scavenging effect of scutellarein on superoxide anions is slight, its other anti-oxidative effects are satisfactory in cell-free systems. As shown in Appendix A–E, scutellarein possessed significant ferric-reducing ability (50 μM of scutellarein equivalent to about 175 μM of Trolox), marked scavenging activity for DPPH (IC_50_ = 20.71 ± 0.66 μM) and hydroxyl radicals (IC_50_ = 31.76 ± 2.53 μM), as well as the effective inhibition of MDA formation induced by Fe^2+^ (IC_50_ = 52.37 ± 9.69 μM).

In response to LPS, epithelial cells can also produce a series of inflammatory cytokines. IL-6, promptly and transiently produced in response to infections and tissue injuries, contributes to host defense through the stimulation of acute phase responses, hematopoiesis, and immune reactions [39]. As such a multi-functional cytokine, it plays a vital role in the pathological process of infectious respiratory diseases, including COVID-19 [40]. CCL2, a critical chemokine, can recruit monocytes to inflammatory sites within the lung [41]. It is also implicated in the process of airway remodeling [42], which contributes to the development of chronic lung diseases. CXCL8, predominantly produced by lung epithelial cells, is the main chemokine responsible for neutrophil recruitment [43,44]. It has been associated with the pathogenesis of various lung diseases, including ARDS, chronic obstructive pulmonary disease, and asthma [45]. In our study, we investigated the effects of scutellarein on the production of IL-6, CXCL8, and CCL2 in LPS-activated human epithelial cells (BEAS-2B). The results demonstrated that scutellarein exhibited significantly suppressive effects on both the transcriptional and translational levels of these inflammatory cytokines (Figure 3 and Figure 4).

LPS-triggered inflammatory responses are primarily governed by NF-κB and AP-1 signaling. In quiescent cells, NF-κB exists as an inert heterodimer composed of two subunits (p50 and p65) and binds with the inhibitor of NF-κB (IκBα) to create a complex residing in the cytoplasm. Upon LPS stimulation, the activated IκB kinase (IKK) rapidly phosphorylates IκBα, leading to its ubiquitination and subsequent degradation. This process culminates in the liberation of NF-κB (p50/p65 heterodimer). Subsequently, liberated NF-κB promptly moves into the nucleus and attaches to κB sites situated within the promoters/enhancers of inflammation-related genes [46,47]. Our findings demonstrated that scutellarein prevented IκBα’s phosphorylation and degradation, thus consequently restraining p65’s migration into the nucleus and inhibiting p65–DNA binding (Figure 5). AP-1 is another vital transcription factor that is subject to the regulation of the MAPK family. The phosphorylation of any MAPK member can activate AP-1 signaling [48]. However, scutellarein did not affect the phosphorylation of MAPKs, indicating its selective inhibition of NF-κB rather than AP-1 signaling (Figure 5 and Figure 6).

Based on the in vitro anti-inflammatory action of scutellarein, we also investigated its in vivo effect. Indeed, scutellarein not only reduced the levels of IL-6 and CCL2 in the BALF of LPS-induced ALI mice (Figure 7A,B) but also significantly decreased the level of TNF-α (Figure 7C), another essential pro-inflammatory cytokine that could not be suppressed by scutellarein in LPS-stimulated BEAS-2B cells. Given that the CXCL8 gene is absent in rodents and CXCL1 in rodents serves similar biological functions to CXCL8 in humans [49], we also determined the effect of scutellarein on CXCL1 in the BALF. However, it only slightly affected CXCL1. In fact, besides epithelial cells, a wide range of other cell types also participate in the initiation and development of ALI, such as neutrophils, macrophages, endothelial cells, and pneumocytes [50]. Thus, the pro-inflammatory cytokines in the BALF may be derived from various cells. Therefore, it is not surprising that the in vitro and in vivo effects of scutellarein on some cytokines are inconsistent. Besides reducing the levels of pro-inflammatory cytokines in the BALF, scutellarein could also alleviate LPS-caused pathological changes in the lung, such as septal thickening, interstitial edema, alveolar hemorrhage, and pronounced neutrophilic infiltration (Figure 7D,E and Figure 8).

## 5. Conclusions

The present study investigated, for the first time, the anti-oxidative and anti-inflammatory activities of scutellarein in LPS-stimulated epithelial cells and LPS-induced ALI. In LPS-stimulated BEAS-2B cells, it not only could reduce intracellular ROS accumulation by inhibiting the activation of NADPH oxidases but could also decrease the transcription and translation of pro-inflammatory cytokines (IL-6, CXCL8, and CCL2) through suppressing the activation of NF-κB signaling (Figure 9). In LPS-induced ALI mice, scutellarein lowered the levels of pro-inflammatory cytokines in the BALF and relieved the pathologic changes in the lung. These findings suggest that scutellarein may be a beneficial agent for the treatment of infectious pneumonia. Further studies are warranted to identify the potential target for the effects of scutellarein, as well as clarify the underlying mechanism of its suppressive action on the infiltration of neutrophils whose extreme accumulation can result in excessive lung injury associated with inflammation and oxidative stress [51].

## Figures and Tables

**Figure 1 antioxidants-13-00710-f001:**
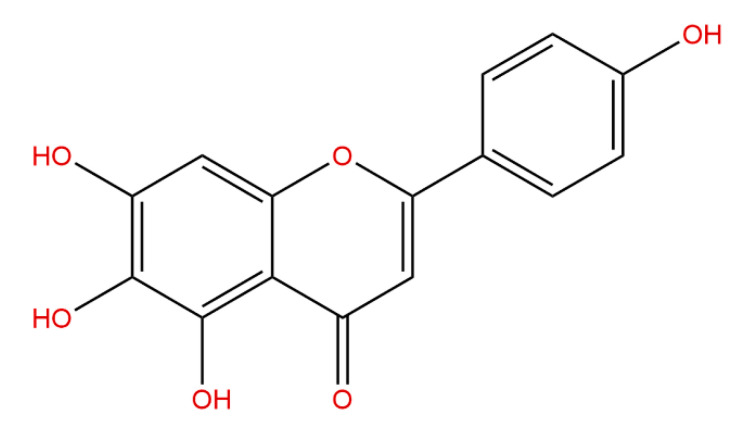
Chemical structure of scutellarein.

**Figure 2 antioxidants-13-00710-f002:**
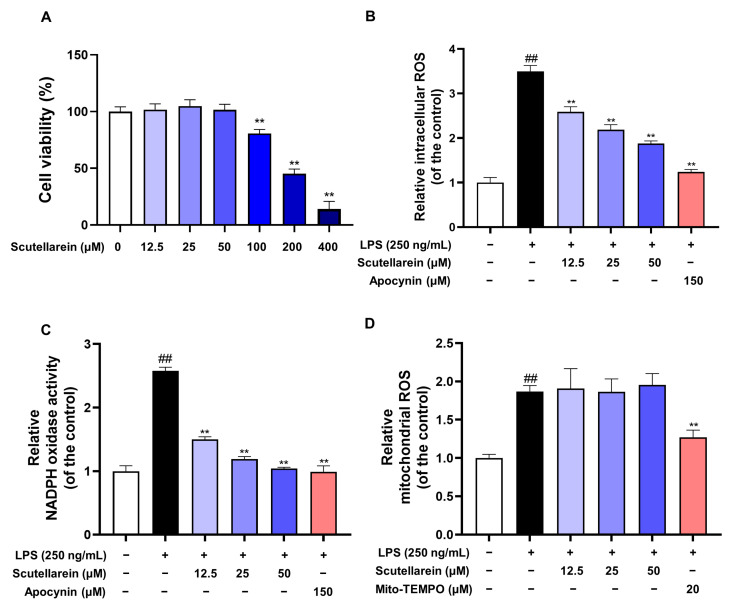
Anti-oxidative activity of scutellarein in LPS-activated BEAS-2B cells (n = 3). (**A**) Effect of scutellarein on the viability of BEAS-2B cells. (**B**–**D**) Effects of scutellarein on the intracellular ROS levels (**B**), NADPH oxidative activity (**C**), and mitochondrial ROS (**D**) levels in LPS-activated BEAS-2B cells. Apocynin and Mito-TEMPO served as the positive controls. ^##^
*p* < 0.01 vs. negative control; ** *p* < 0.01 vs. LPS alone.

**Figure 3 antioxidants-13-00710-f003:**
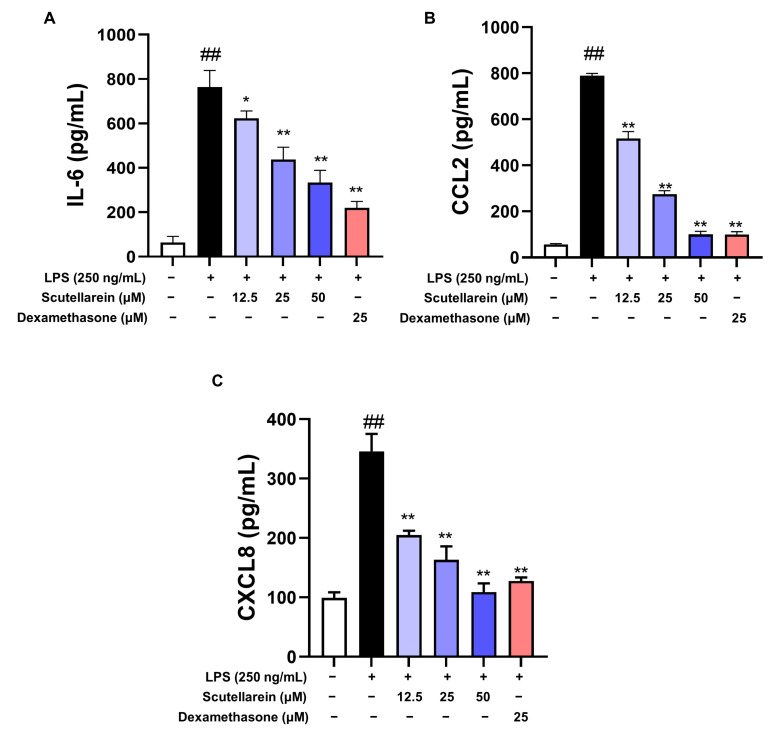
Effects of scutellarein on the levels of the supernatant pro-inflammatory mediators ((**A**) IL-6; (**B**), CCL2; (**C**), CXCL8) of LPS-stimulated BEAS-2B cells (n = 3). Dexamethasone served as a positive control. ^##^
*p* < 0.01 vs. negative control; * *p* < 0.05 and ** *p* < 0.01 vs. LPS alone.

**Figure 4 antioxidants-13-00710-f004:**
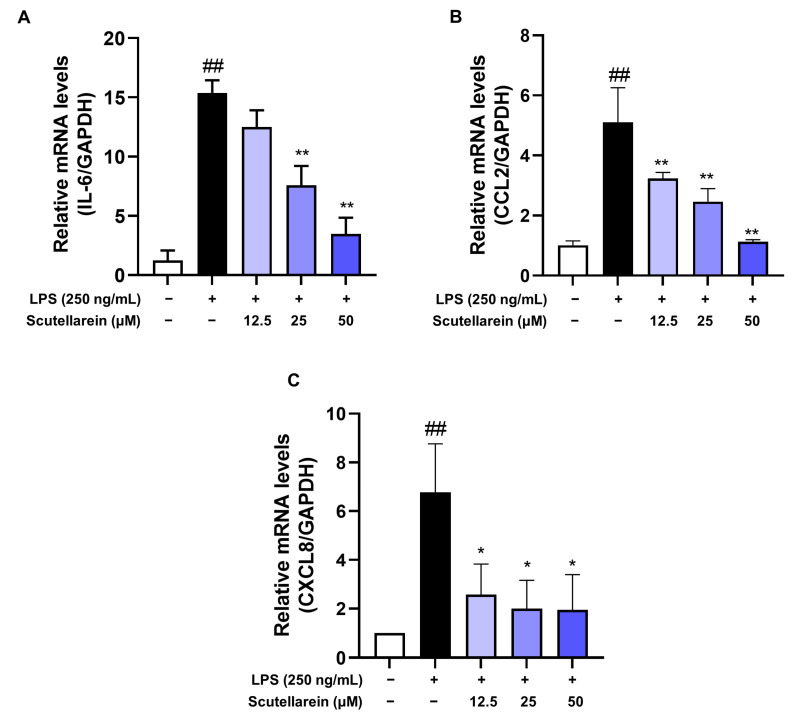
Effects of scutellarein on the mRNA levels of IL-6 (**A**), CCL2 (**B**), and CXCL8 (**C**) (n = 3). ^##^
*p* < 0.01 vs. negative control; * *p* < 0.05 and ** *p* < 0.01 vs. LPS alone.

**Figure 5 antioxidants-13-00710-f005:**
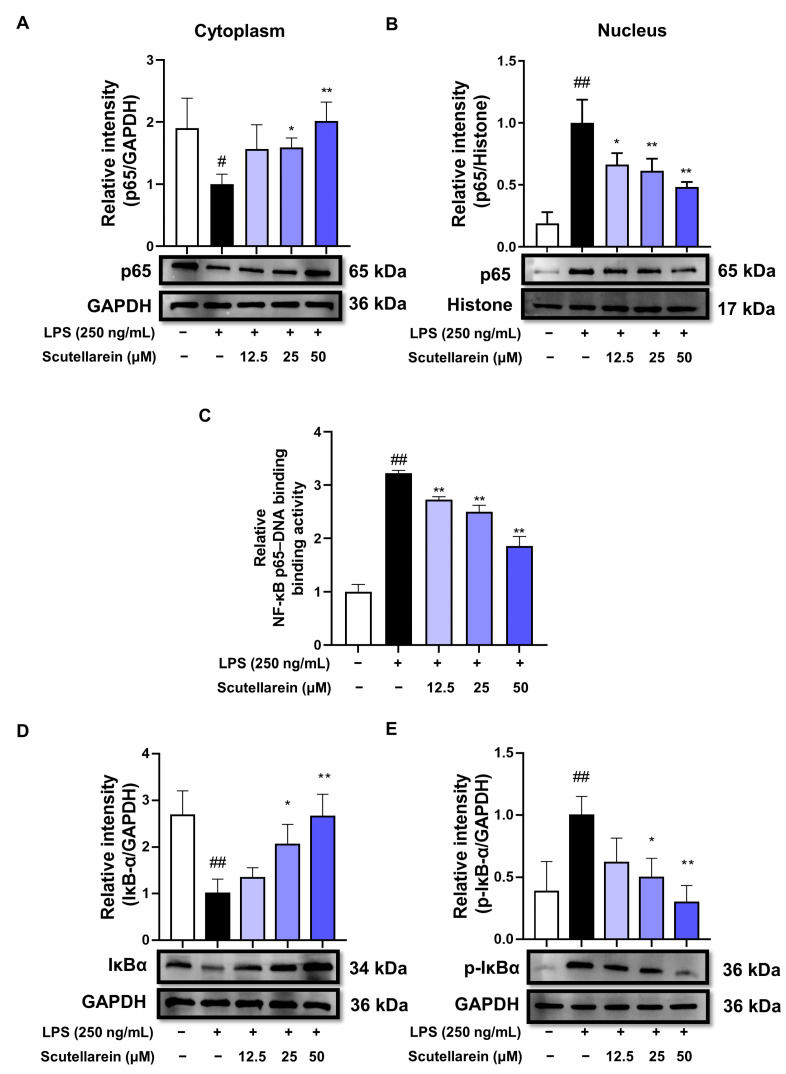
Effect of scutellarein on NF-κB signaling in LPS-activated BEAS-2B cells (n = 3). (**A**,**B**) Effect of scutellarein on LPS-induced p65 nuclear translocation. ^#^
*p* < 0.05 and ^##^
*p* < 0.01 vs. negative control; * *p* < 0.05 and ** *p* < 0.01 vs. LPS alone. (**C**) Effect of scutellarein on LPS-induced p65–DNA binding. ^##^
*p* < 0.01 vs. negative control; ** *p* < 0.01 vs. LPS alone. (**D**,**E**) Effects of scutellarein on LPS-induced IκBα degradation and phosphorylation. ^##^
*p* < 0.01 vs. negative control; * *p* < 0.05 and ** *p* < 0.01 vs. LPS alone.

**Figure 6 antioxidants-13-00710-f006:**
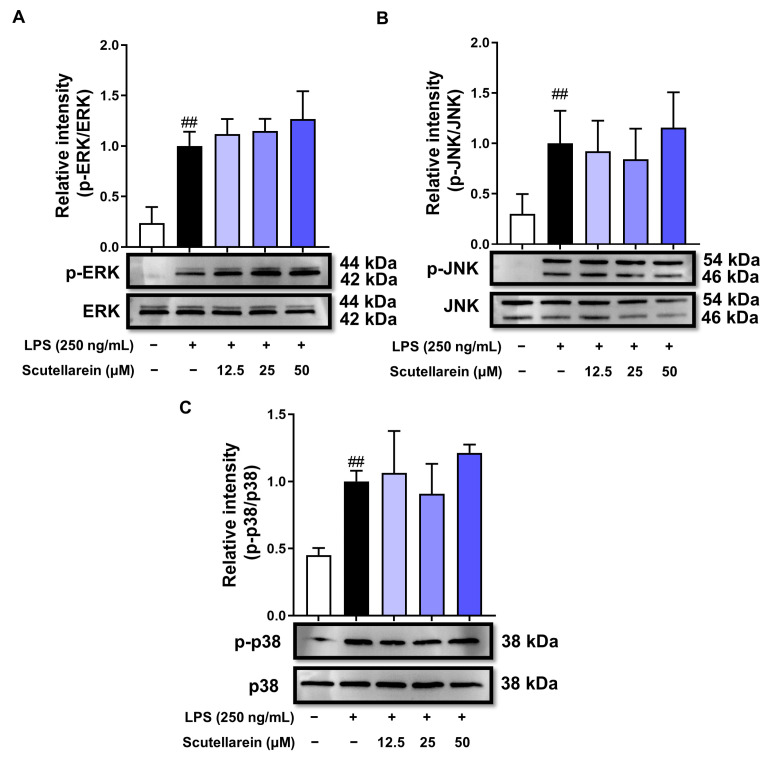
Effect of scutellarein on AP-1 signaling in LPS-activated BEAS-2B cells (n = 3). (**A**–**C**) Effects of scutellarein on the LPS-induced phosphorylation of MAPK family members [ERK (**A**), JNK (**B**), and p38 (**C**)]. ^##^
*p* < 0.01 vs. negative control.

**Figure 7 antioxidants-13-00710-f007:**
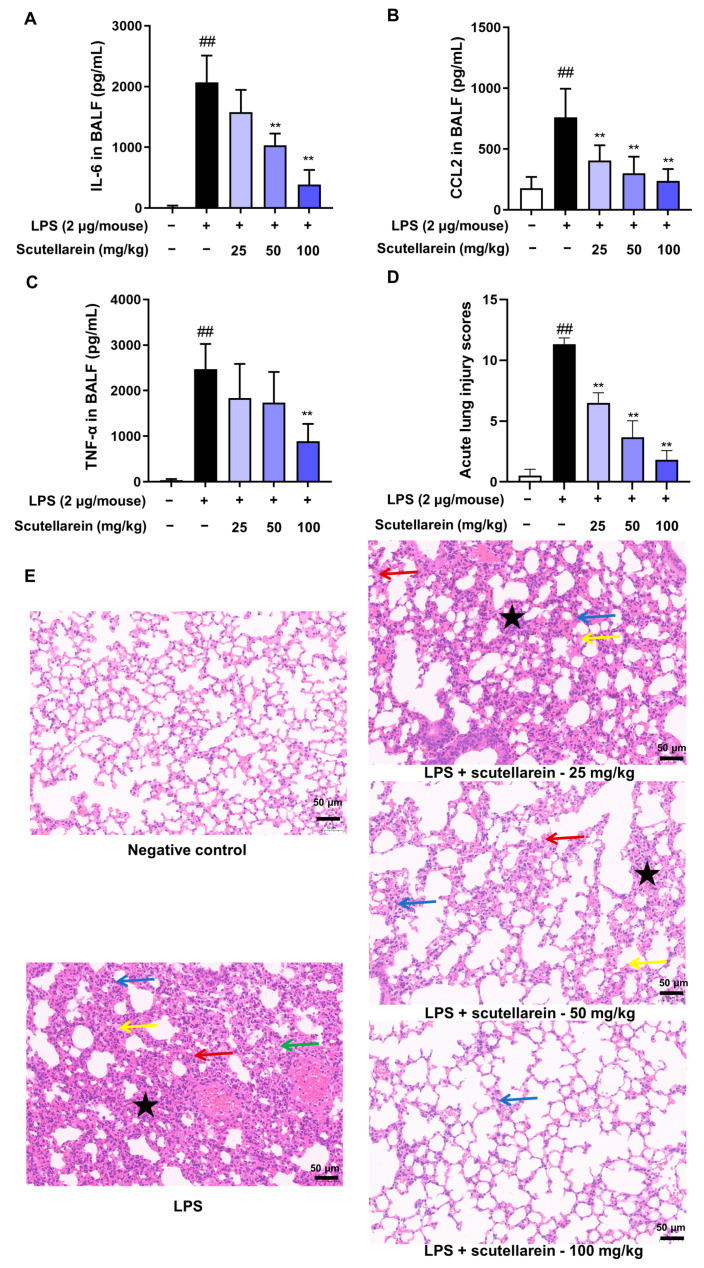
Effects of scutellarein on LPS-induced ALI mice. ALI was induced via the intratracheal delivery of LPS (2 μg/20 μL/mouse) using a mouse microsprayer aerosolizer. (**A**–**C**) Effects of scutellarein on the levels of IL-6 (**A**), CCL2 (**B**), and TNF-α (**C**) in BALF (n = 8). (**D**) Lung injury scores were calculated according to the predetermined criteria. ^##^
*p* < 0.01 vs. negative control; ** *p* < 0.01 vs. LPS alone. (**E**) Representative images of H and E staining in lung tissues (200×). The green arrows in the figure point to the edema condition. The blue arrows point to alveolar and interstitial inflammation. The red arrows point to alveolar and interstitial hemorrhage. The yellow arrows point to hyaline membrane formation. The black pentagrams point to atelectasis. Scale bars, 50 µm.

**Figure 8 antioxidants-13-00710-f008:**
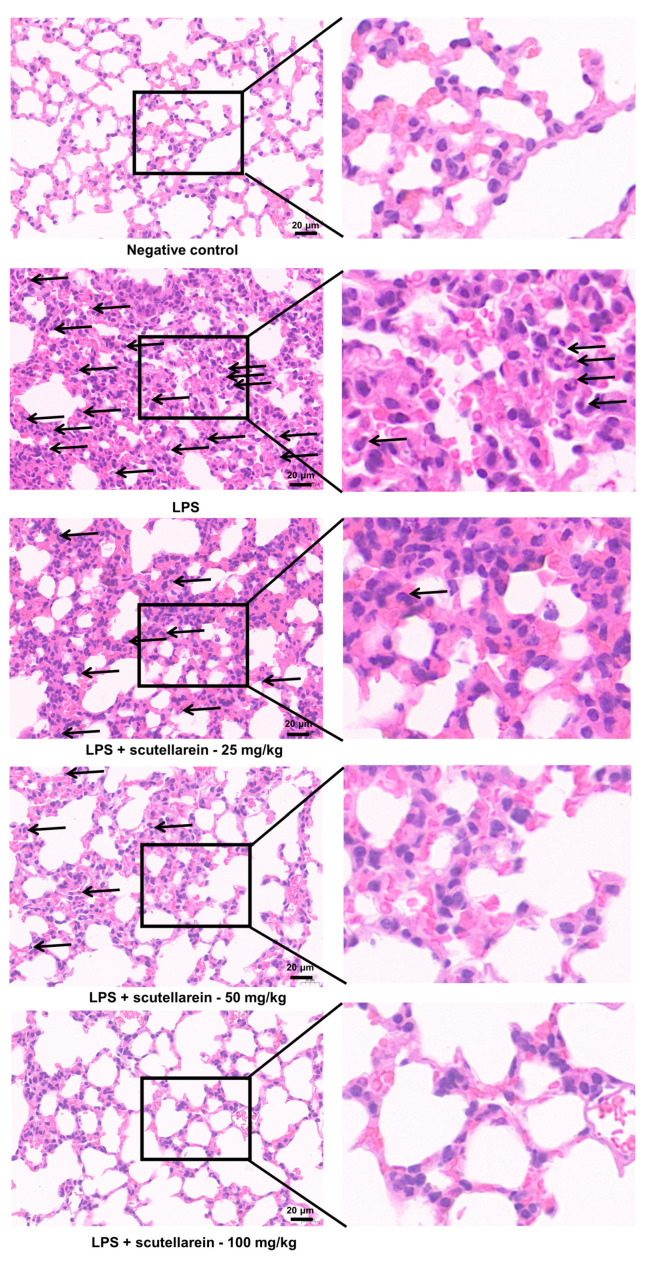
Effect of scutellarein on neutrophilic infiltration in LPS-induced ALI mice. Representative images of H and E staining in lung tissues (400×). The black arrows point to neutrophils. Scale bars, 20 µm.

**Figure 9 antioxidants-13-00710-f009:**
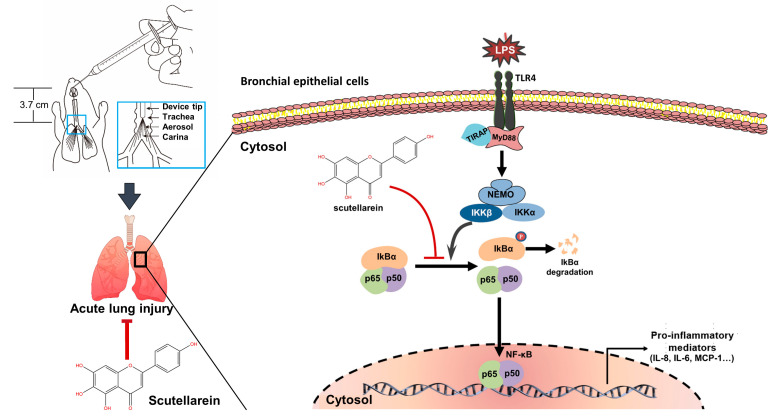
Schematic diagram illustrating the anti-inflammatory effects of scutellarein in the LPS-activated bronchial epithelial cells of acute lung injury mice. Black arrow means positive regulation. Red “T” means inhibition.

**Table 1 antioxidants-13-00710-t001:** Sequences of primers used in human IL-6, CCL2, CXCL8, and GAPDH.

Gene	Accession Number	Strand	Primer Sequence (5′ to 3′)
*IL-6*	XM_054358145	Forward	GAGGAGACTTGCCTGGTGAAA
		Reverse	TTGCATCCCTGAGTTGTCCA
*CCL2*	NM_002982	Forward	TCAAACTGAAGCTCGCACTCT
		Reverse	GGGGCATTGATTGCATCTGG
*CXCL8*	NM_000584	Forward	CTGGCCGTGGCTCTCTTG
		Reverse	TTAGCACTCCTTGGCAAAACTG
*GAPDH*	NM_001256799	Forward	CTCAACTACATGGCTGAGAACG
		Reverse	CATGACGAACATGGGGGCAT

## Data Availability

The original contributions presented in the study are included in the article/Appendix A, further inquiries can be directed to the corresponding author.

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
