# Peer review of "Scutellarein Suppresses the Production of ROS and Inflammatory Mediators of LPS-Activated Bronchial Epithelial Cells and Attenuates Acute Lung Injury in Mice"

_antioxidants, 2024, doi:10.3390/antiox13060710_

Round 1

Reviewer 1 Report

None

None

Author Response

Dear the reviewer,

Thank you for the kind assistance and the critical review to improve our manuscript entitled "Scutellarein suppresses the production of ROS and inflammatory mediators of LPS-activated bronchial epithelial cells and attenuates acute lung injury in mice" (No. antioxidants-3020277). We have carefully studied your comment and have made substantial revisions. An editable version of the article has been uploaded. In this revised version, all changes were highlighted by yellow area.

The article mentions how in the ICR mouse with the acute lung injury model, the scutellarein inhibits neutrophil infiltration, but these results are not observed, please include them.

Response: Thanks for the helpful suggestion. We have marked the neutrophils by black arrows under a 400x magnification view (page 13-14, Figure 8).

We sincerely hope the revised manuscript can be suitable for the publication.

Sincerely,

Yuan Gao & Yun Qi

Reviewer 2 Report

The research paper manuscript by Ximeng Li et al. test the Scutellarein function in vivo and in vitro. The results show Scutellarein can reduce intracellular ROS accumulation by inhibiting NADPH oxidases and downregulates the transcription and translation of pro-inflammatory cytokines (IL-6, CCL2, CXCL8) in LPS-activated BEAS-2B cells. At same time, they show Scutellarein inhibits IκBα phosphorylation and degradation, preventing p65 translocation to the nucleus and subsequent NF-κB-regulated gene transcription, without affecting AP-1 signaling. In LPS-induced ALI mice, scutellarein decreases IL-6, CCL2, and TNF-α levels in bronchoalveolar lavage fluid, reduces lung injury, and inhibits neutrophil infiltration. Their study suggests that scutellarein may be an effective agent for treating infectious pneumonia due to its anti-oxidative and anti-inflammatory activities.

There are some points that author should be considered:

1.      Some format of this paper should be correct: Page4, part 2.8 and part 2.9., Fonts are inconsistent with other parts.

2.      For the in vivo study, do you have any data about what’s the pharmaceutical kinetic looks like of Scutellarein in the mice body? What’s the concentration in mice serum?

3.      According to the results, it seems like the Scutellarein has a positive effect for the ALI mice. It would be good to depict a diagram to show how the Scutellarein play a role in anti-inflammation.

  For the in vivo study, do you have any data about what’s the pharmaceutical kinetic looks like of Scutellarein in the mice body? What’s the concentration in mice serum?

According to the results, it seems like the Scutellarein has a positive effect for the ALI mice. It would be good to depict a diagram to show how the Scutellarein play a role in anti-inflammation.

Author Response

Dear the reviewer,

Thank you for the kind assistance and the critical review to improve our manuscript entitled "Scutellarein suppresses the production of ROS and inflammatory mediators of LPS-activated bronchial epithelial cells and attenuates acute lung injury in mice" (No. antioxidants-3020277). We have carefully studied all comments and have made substantial revisions. An editable version of the article has been uploaded. In this revised version, all changes were highlighted by yellow area.

(1) Some format of this paper should be correct: Page 4, part 2.8 and part 2.9., Fonts are inconsistent with other parts.

Response: Thank you so much for the careful check. We have corrected this clerical error in the revised manuscript (page 4, line 147-152).

(2) For the in vivo study, do you have any data about what’s the pharmaceutical kinetic looks like of Scutellarein in the mice body? What’s the concentration in mice serum?

Response: Pharmacokinetic studies are usually conducted in rats, even dogs, which possess sufficient blood volume for determination. Since the pharmacokinetics data of scutellarein have been well studied, we didn't design this content in the present study. It has been demonstrated that the Cmax value of scutellarein was 3.44 μg/mL at the Tmax of 0.5 h, and the T1/2 of 11.60 h, as well as AUC0–∞ of 26.25 mg • h/L in rat plasma. In human plasma, scutellarein was associated with the Cmax of 155 ng/mL and AUC0–∞ of 1104 ng • h/mL at the Tmax of 7.3 h and T1/2 of 3.1 h. The absolute bioavailability of scutellarein following the oral administration to dogs is very low at about 0.40% [Scutellarein: a review of chemistry and pharmacology. J Pharm Pharmacol. 2024 Apr 5: rgae039. doi: 10.1093/jpp/rgae039].

(3) According to the results, it seems like the Scutellarein has a positive effect for the ALI mice. It would be good to depict a diagram to show how the Scutellarein play a role in anti-inflammation.

Response: Thank you for the suggestion. We have depicted the diagram in the revise manuscript (page 16, Figure 9).

(4) It is necessary to include study limitations in the discussion.

Response: Thanks for the valuable suggestion. We have included study limitations in the conclusion (page 15, line 401-404).

We sincerely hope the revised manuscript can be suitable for the publication.

Sincerely,

Yuan Gao & Yun Qi

Reviewer 3 Report

The inclusion of the in vitro data in the manuscript ensure a clear analysis of the physiological effect of the scutellarein compound on the NF-kappaB and not the MAPKs/AP-1 pathway. The authors also include a clear concentration range of the compound in the in vitro data. 

The in vivo data are less rigorously investigated. They include the protein level analysis in the BALF and some histological images. The latter are not sufficient in my opinion (only representative images without a quantification). The authors should include a better analysis (e.g., acute lung inury score) in order to prove this in a quantitative manner. 

Line 63-64: what is meant with heat-clearing and dampness-reducing?

Line 190: how is the position (in the lumen of the trachea) of the aerosolizer checked and the intratracheal delivery proven? 

Author Response

Dear the reviewer,

Thank you for the kind assistance and the critical review to improve our manuscript entitled "Scutellarein suppresses the production of ROS and inflammatory mediators of LPS-activated bronchial epithelial cells and attenuates acute lung injury in mice" (No. antioxidants-3020277). We have carefully studied all comments and have made substantial revisions. An editable version of the article has been uploaded. In this revised version, all changes were highlighted by yellow area.

(1) The inclusion of the in vitro data in the manuscript ensure a clear analysis of the physiological effect of the scutellarein compound on the NF-kappaB and not the MAPKs/AP-1 pathway. The authors also include a clear concentration range of the compound in the in vitro data.

The in vivo data are less rigorously investigated. They include the protein level analysis in the BALF and some histological images. The latter are not sufficient in my opinion (only representative images without a quantification). The authors should include a better analysis (e.g., acute lung injury score) in order to prove this in a quantitative manner.

Response: Thank you for the valuable suggestion. We have included the acute lung injury score in the revised manuscript to provide a more quantitative analysis (page 5, line 198-204; page 12-13, Figure 7D).

(2) Line 63-64: what is meant with heat-clearing and dampness-reducing?

Response: "Heat-clearing" and "dampness-reducing" are terminology of traditional Chinese medicine (TCM) theory without any corresponding English explanation. To avoid potential misunderstanding by the readers, we have removed these terms from the text. This change will not affect the overall comprehension of the manuscript.

(3) Line 190: how is the position (in the lumen of the trachea) of the aerosolizer checked and the intratracheal delivery proven?

Response: Thanks for your reminder. We have illustrated the position and depth of the intratracheal delivery in Figure 9 (page 16).

(4) It is necessary to include study limitations in the discussion.

Response: Thanks for the valuable suggestion. We have included study limitations in the conclusion (page 15, line 401-404).

We sincerely hope the revised manuscript can be suitable for the publication.

Sincerely,

Yuan Gao & Yun Qi

Round 2

Reviewer 2 Report

The author response the question properly and no more comments.    

The author response the question properly and no more comments.    

Reviewer 3 Report

The authors have addressed my major comments and have included the assessment of the lung injury score.

No other comments